# Evaluation of an On-Site Disaster Medical Management Course in Nepal

**DOI:** 10.3390/healthcare12131308

**Published:** 2024-06-30

**Authors:** Joy Li-Juan Quah, Joost Bierens, Venkataraman Anantharaman

**Affiliations:** 1Department of Emergency Medicine, Singapore General Hospital, Emergency Medicine Academic Clinical Programme, Duke-NUS Academic Medical Centre, 1 Outram Road, Singapore 169608, Singapore; joy.quah.l.j@singhealth.com.sg; 2Research Group Emergency and Disaster Medicine, Vrije Universiteit Brussel, Laarbeeklaan 103, 1090 Brussels, Belgium; jbierens@euronet.nl

**Keywords:** disasters, disaster medicine, disaster planning, training, workshop

## Abstract

The great 2015 Nepal earthquake of magnitude 7.6 killed about 9000 people. To better ensure a more coordinated disaster response, a Basic On-Site Disaster Medical Support (BOS-DMS) course was designed in 2017. This study evaluates the effectiveness of the BOS-DM course. The course was conducted twice and attended by 135 participants, of whom 113 (83.7%) answered pre-test and post-test based multiple-choice questions. Qualitative and quantitative feedback was provided by 94 participants (69.6%). Mean test scores for the participants increased from 4.24 ± 1.42 to 6.55 ± 2.16 (*p*-value < 0.0001; paired *t*-test). More than 92.0% of participants felt that the course prepared healthcare workers to manage acute medical situations at a disaster site. Subject knowledge scores increased from 34.8% to 90.2%. A three-day BOS_DMS course has the potential to improve on-site disaster management knowledge. Our study noted that precise scheduling, making attendance compulsory, translating course materials into the local language, inclusion of disaster exercises and training local master trainers can enhance course effectiveness.

## 1. Introduction

The United Nations lists Nepal as the 11th-most earthquake-prone country in the world [1]. The last major earthquake was in April 2015 with a magnitude of 7.6 as recorded by Nepal’s National Seismological Centre, which was followed by more than 300 aftershocks of magnitude greater than 4.0. The official death toll was estimated at 9000 lives. Resultant damages and losses were estimated at USD 7065 million [1].

Within hours, the Government of Nepal requested for international assistance as local services were being overwhelmed. Emergency relief and humanitarian assistance came from over 60 countries, all deployed through the Nepalese Ministry of Health and Population (MOHP)’s Health Emergency Operations Centre (HEOC) [2]. In May 2016, 135 Nepali representatives from community services, academia, the healthcare sector, armed forces, as well as local and national government bodies attended a workshop. The workshop was co-organized by public health specialists from Nepal’s Manmohan Memorial Institute of Health Sciences, University of Sheffield’s (United Kingdom) School of Health and Related Research and local and international non-governmental organizations. One area of improvement that was identified was more immediate and targeted local disaster response coordination [3].

As one of the next steps, in May 2017, a tripartite collaboration between the Government of Nepal, Temasek Foundation International (a philanthropic organization from Singapore), and Singapore Health Services (SingHealth) was established. SingHealth, Singapore’s largest public healthcare service provider, sought to address areas for potential development in disaster-preparedness-related services through interprofessional education [4].

The collaboration resulted in a needs analysis study in Kathmandu conducted by four SingHealth Emergency Physicians in 2017. They identified five areas that require the development of disaster preparedness within the healthcare sector in Nepal. These were community response in disasters, on-site disaster medical management, ambulance support, hospital disaster preparedness and a national framework for disaster coordination.

Nepal had adopted the HOPE (Hospital Preparedness in Emergencies) project in 2005 [5]. HOPE was developed by the United States Agency for International Development as part of its programme for the enhancement of emergency response and has been implemented in a few Asian countries. The focus of the HOPE courses was on the design of health facilities, with specific plans to maximize their ability to withstand incidents that may cause structural damage. The course also included instruction in hospital emergency incident command systems, hospital disaster planning, hospital evacuation, and management of the deceased. There was one lecture on need for on-site care. Financial challenges stood in the way of introducing the course to many healthcare workers there [5].

However, there had been no major effort to work out plans to address immediate on-site medical care at disaster sites because of the perception that the Armed Forces would respond to an incident site within hours of activation [6]. The need to activate the Armed Forces, mobilize their personnel and redeploy them in civilian disaster sites would take some hours and sometimes days, by which time badly injured casualties may not be salvageable [7]. Very few courses on on-site medical support for healthcare professionals were then available. The Basic Disaster Life Support courses are short, of about 7.5 h in duration, and mainly achieve some level of awareness amongst attendees [8]. The Advanced Disaster Life Support Course is a two-day programme with four practical periods that focus on chemical decontamination and information systems and technology and may not be easily widely applicable in Nepal [9]. The University of Hasanuddin in Makassar, Indonesia, conducts regular disaster-site training and multi-agency disaster exercises for their healthcare workers in preparation for local disaster events [10].

To address the need for the development of immediate on-site disaster medical management in Nepal, the Basic On-Site Disaster Medical Support (BOS-DMS) course was created. The objective of this course, which adopted an all-cause disaster management approach, was to enable local healthcare institutions and healthcare workers in the vicinity of a disaster to mobilize their resources to provide coordinated medical assistance at the local disaster site as soon as possible after the onset of the incident as is recently being arranged in a number of other countries [11,12].

### Study Objective

The objective of this study was to determine whether the BOS-DMS course contributed to knowledge improvement of on-site disaster medical management among attending participants.

## 2. Materials and Methods

### 2.1. Development of Course

A three-day workshop-style programme involving didactic and practical break-out sessions with small group facilitation was developed by a group of eight emergency physicians from the SingHealth Duke-National University of Singapore (SingHealth Duke-NUS)’s Academic Medical Centre. The team reviewed the existing disaster literature relevant to the Nepalese environment, created a 179-page course manual (Table of Contents in Table 1) and set up the programme to be taught during a three-day course (Appendix A). United Nations Disaster Risk Reduction (UNDRR) definitions and SORT/START triage systems were used, where applicable [13,14]. The content was specific for a Nepalese audience.

The course was conducted in English, a language common to both instructors from Singapore and participants from Nepal. The course contents were vetted by a group of Nepalese doctors and disaster administrators from MOHP for accuracy and relevance. Teaching materials included slide presentations for lectures, paper handouts for small workgroup break-out sessions and staff aids to enable demonstration and deployment of a disaster-site first-aid post. Up to 14 interactive small workgroup break-out sessions took place during the lectures. Each workgroup consisted of a mix of eight to ten participants, with a SingHealth emergency physician as a facilitator to guide discussion and practice and answer queries. Every workgroup was expected to be able to address and solve near-realistic scenarios.

The practical scenarios used to provide practical experience were as follows:Multiple trauma triage scenarios that may be expected in mass casualty situations.Medical support planning for on-site medical support in an aircrash situation.Practising activation and dispatch of medical teams to a disaster site.Practice setting up a first-aid post and basic simulation using casualty cards.Practical training on disaster-site communications at a first-aid post.Practising command and control of a first-aid post at a disaster site.Practical training on scenarios that may be encountered at the Red, Yellow and Green areas and at the Ambulance point of a first-aid post.Practical training on medical support at disaster sites during floods, landslides and fires.Practical training on decontamination techniques during chemical disasters.Small-group discussions on how to conduct disaster-site exercises.

These scenarios were relevant to the local setting and addressed lessons covered in the lectures. Presentations by each workgroup to the whole class provided opportunities for larger group discussions and the sharing of personal experiences.

### 2.2. Course Participants

The first course was conducted for three days from 2 to 4 May 2018. The course participants were from a purposive sample of healthcare workers in Nepal. Criteria for selection of participants included those already identified to have potential to become master trainers by their institutions for the BOS-DMS course and able to speak, read and write in the English language. Based on these criteria, the hospital senior management and local community leaders selected the participants. Selected doctors were senior physicians/surgeons already in practice for at least five years and had completed their specialist post-graduate training. Selected nurses and administrators were to have been in practice for at least five years and working in senior appointments or as educators.

The second course was condensed into a two-day programme and ran from 26 to 27 November 2018 at the request of local health authorities, owing to the time constraints of the participants. The content of the entire course manual was still covered.

### 2.3. Study Design and Course Assessments

This was a quasi-experimental design with pre- and post-tests immediately before and after the conduct of the BOS-DMS course. Each participant had to perform both assessments. Both tests included multiple-choice questions, with one mark awarded for a correct answer. The maximum total score was ten marks. There was no discussion of the test questions or correct answers after the pre-test. After the post-test, the correct answers were revealed with discussion. The multiple-choice test was developed to guide learning and test important core concepts covered during the course. The test questions had been used in previous courses on disaster readiness conducted by the faculty but were suitably modified for use in Nepal [10]. Areas covered included disaster definitions and terminology (Q1), disaster emergency services and their roles (Q2), the organization of medical services at a disaster site (Q3), healthcare system coordination in disasters (Q4), principles of hospital-level medical support (Q5, Q6, Q7), on-site disaster triage (Q8), organization and components of a first-aid post (Q9) and casualty evacuation principles (Q10). After the full course was finished, feedback was obtained to evaluate the effectiveness of the course and the trainers, achievement of learning outcomes and the facilities provided. The feedback questionnaire had also been used by the course faculty for previous training courses conducted in other countries. Participants had to rate, on a 3-point Likert scale, whether they agreed, were neutral or disagreed with given statements. There was also provision for qualitative feedback. The participants provided these inputs on the feedback forms manually. These were then transferred to an electronic database for analysis.

### 2.4. Statistical Analysis

Overall mean test scores of the ten questions with standard deviations were calculated for the participants. The mean imputation statistical method was used to fill in missing corresponding pre-test or post-test data for participants who only submitted either test scripts, using the mean test score of participants for that particular test and course. Pre-test and post-test results before and after imputation were compared using paired *t*-test. A *p*-value < 0.05 would suggest a statistically significant difference between the pre-test and post-test mean scores for the particular question.

As each question was formulated to test a certain disaster topic, descriptive analysis was also carried out to help understand which topics demonstrated the best, moderate or least improvement after the programme. For the feedback portion, descriptive analysis was carried out and presented as a percentage of participants who agreed, were neutral or disagreed with each statement. Fisher’s exact test was used to compare between the three categorical variables. A *p*-value < 0.050 would suggest a statistically significant finding for that statement.

### 2.5. Ethical Considerations

Ethics exemption was obtained from the SingHealth Institutional Review Board on 20 December 2018. As part of the memorandum of understanding between SingHealth and MOHP, Nepal, consent had been obtained from all participants once they agreed to be part of the training programme with assurance of confidentiality. Collection and protection of data was in accordance with the Personal Data Protection Act under Singapore Law, and solely for programme-related activities and evaluation [15].

## 3. Results

### 3.1. Participant Characteristics

A total of 14 healthcare institutions and two non-governmental organizations took part in the two courses with 135 participants representing a variety of professions and affiliations (Table 2, Appendix A). Of these, 65 were male and 70 were female.

### 3.2. Mean Test Scores (Table 3)

A total of 107 participants took the pre-test (79.3%), 93 (68.9%) the post-test and 87 (64.4%) took both. For the first course, there were 55 registered participants of whom 43 (78.2%) took the pre-test, 34 (61.8%) the post-test and 32 (58.2%) both. For the second course, there were 80 registered participants; 64 (80.0%) took the pre-test, 59 (73.8%) the post-test and 55 (68.8%) both.

Before imputation, the mean test scores of the 87 participants who took both tests increased from 4.38 ± 1.48 to 6.40 ± 2.36 (*p* < 0.001). For the first course, the mean test scores for the 32 who took both increased from 4.19 ± 1.09 to 7.38 ± 1.91 (*p* < 0.001). For the second course, the mean test scores for the 55 who took both increased from 4.49 ± 1.68 to 5.84 ± 2.42 (*p* < 0.001).

After imputation, the overall mean test scores increased from 4.24 ± 1.42 pre-test to 6.55 ± 2.16 post-test (*p* < 0.001). For the first course, the mean test score increased from 3.95 ± 1.13 to 7.63 ± 1.57 (*p* < 0.001). For the second course, the mean test score increased from 4.42 ± 1.56 to 5.85 ± 2.21 (*p* < 0.001). Table 3 provides a question-by-question analysis of the results. The mean imputation statistical method demonstrated similar changes in the test scores as for before imputation, when applied to all those who took the tests.

For the doctors attending the courses, the mean test scores after imputation increased from 5.42 ± 1.77 to 7.63 ± 2.04 (*p* < 0.001). For the nurses, the scores increased from 4.11 ± 1.22 to 6.30 ± 2.27 (*p* < 0.001).

**Table 3 healthcare-12-01308-t003:** Pre-test and post-test scores after imputation by question for both May 2018 and November 2018 courses.

	Topic	Question	May 2018 Correct Responses (%)			November 2018 Correct Responses (%)		
			Pre-Test *n* = 43	Post-Test *n* = 34	*p*-Value	Pre-Test *n* = 64	Post-Test *n* = 59	*p*-Value
Q1	Disaster Definitions and Terminology	What are the four phases in the disaster management cycle?	18.6	88.2	<0.001	26.6	83.1	<0.001
Q2	Disaster Emergency Services and Their Roles	What are the designated roles of the following agencies in a disaster? Rescue Services, Police, Medical Services, Social Services.	58.1	61.8	0.79	67.2	37.3	0.002
Q3	Organization of Medical Services at a Disaster Site	Where and how do medical teams usually operate at the disaster site?	11.6	94.1	<0.001	26.6	42.4	0.015
Q4	Coordination of Disaster Medical Services	Identifying true statements on activation and coordination of medical service agencies during a disaster	34.9	58.8	0.11	31.3	50.8	0.06
Q5	Hospital-Level Medical Support	What are some of the correct actions taken by public hospitals during civil disasters?	88.4	91.2	1.00	87.5	88.1	0.71
Q6	Hospital-Level Medical Support	What are some appropriate arrangements to be made in the hospital during a civil disaster?	30.2	70.6	<0.001	32.8	37.3	0.62
Q7	Hospital-Level Medical Support	What are the key areas of the hospital during a disaster?	90.7	100.0	0.16	95.3	93.2	0.71
Q8	Triage in Disasters	What are correct triage decisions when triaging casualties with specific injuries during a disaster?	46.5	82.4	<0.010	48.4	49.2	0.83
Q9	Triage in Disasters	What are the appropriate treatment decisions to be taken at a disaster-site first-aid post?	11.6	58.8	<0.001	20.3	57.6	<0.05
Q10	Organization and Components of a First-Aid Post	What are correct casualty evacuation principles when sending casualties from the first-aid post to the appropriate hospital?	4.7	67.6	<0.001	6.3	45.8	<0.001

Prior to the start of the training, the participants performed best in the question related to hospital-level medical support (Q7; 90.7%). After the course, this remained the best score. The areas of significant improvements were in disaster definitions and terminology (Q1: +61.6%; *p* < 0.001), organization of medical services at a disaster site (Q3: +40.7%; *p* < 0.001), triage in disasters (Q9: +41.2%; *p* < 0.001) and organization and components of a first-aid post (Q10: +48.3%; *p* < 0.001). The score of the question on disaster emergency services and their roles decreased significantly (Q2: −17.3%; *p* = 0.016), especially for the second course.

In general, the feedback was positive for the course with 92.4% agreeing that the programme had achieved its stated objective of imparting knowledge relating to on-site medical support during a disaster and 93.5% agreeing that they will be able to apply the knowledge and skills to their work. Good knowledge of the subject matter after the workshop increased from 34.8% to 90.2%. Table 4 describes the feedback for both courses. Of note, the condensed second course received less favourable feedback for course delivery, course materials and trainers.

Most participants emphasized on the importance of disaster management courses with comments such as—“such training should be provided to all medical staff”, “this is a crucial training to develop skilled manpower”, “all medical personnel should know about disaster-site medical support for disaster preparedness” and “such programs should be held as regular basic and throughout the country”. The andragogic format of the workshop was also highlighted, with comments like “the course is learner-centered, well-explained and mixed with practical exercise” and requests for even more interactive modules, “…focus more over practical activities and role play, that might be most appropriate and entertaining. Rather than power point presentation”, “focus more on practical activities” and “give enough time for participants to (be) involved in activities”.

The most common constructive feedback of the programme involved the choice of language medium for the course—“you may need a language translator in the next session to avoid language barriers and have better communication with the local people”, “in our context, Nepal, in this training, many participants had language problems, usually the older ones. So, it would have been more fruitful if there were some trainers knowing the local language”. There were also suggestions for additional disaster drills and exercises, such as “suggest drill/simulation of disaster site” and “include simulation of disaster site organization”. Specifically, from the participants of the condensed two-day course, there were several comments on its duration: “duration of the course should be longer”, “the duration of the course was short” and “training should be three days, instead of two days”.

## 4. Discussion

This was the first before-and-after study comparing knowledge scores in disaster medical training in Nepal. In 2017, a systemic review concluded that interprofessional education in global health care benefits from the implementation of cross-border collaboration. The benefits listed include improving the strength and skills of health workers and enhancing the region’s capacity to respond to health security issues [16]. This study confirms that a jointly prepared course is indeed able to increase the knowledge of on-site disaster medical management by the Nepalese medical community

The finding in our study of low pre-test scores in key areas of disaster management, such as disaster definitions, disaster site organization, coordination of medical services, triage and organization of the first-aid post, is not surprising and has been noted in similar studies around the world. This reflects the lack of coverage of disaster preparedness education in many jurisdictions [17,18,19,20]. The better scores in the questions on hospital-level medical support could be expected because the participants work in hospitals and have attained some seniority in their jobs. Some may have been trained in other disaster preparation courses such as the HOPE course. We did not ascertain who amongst the participants had undergone such previous training [21].

The decrease in scores in the question on the role of disaster agencies was mainly observed in the results of the second course.

We identified two areas which can be improved. Firstly, the second course was condensed into two days owing to unforeseen scheduling conflicts between emergency physicians from Singapore and participants from Nepal. This was suboptimal for course delivery and resulted in longer time spent at the course venue each day, frequent interruptions to some of the participants to answer urgent calls from their work places and shorter time available for each topic. The poorer post-test results and course feedback were indicative of this. Furthermore, some participants either came late or left early as attendance was not made compulsory. These tie in with the recommendation from the WHO Framework for Interprofessional Education and Collaborative Practice that compulsory attendance and flexible scheduling tend to improve outcomes [22,23]. Better liaison efforts may be able to mitigate this challenge in the future. In the age of rising pandemics, one may also need to consider internet, web conferencing tools and multi-user virtual environments to conduct interprofessional collaboration effectively [24,25,26].

Secondly, language was cited by the participants as a barrier to the effectiveness of course delivery. The concern over language as a barrier to effective teaching was considered at the start of the collaboration; local senior management was consulted about this and course faculty were reminded of the need to be clear in their diction. Use of translators during the course would have severely prolonged course duration, and scarcity of time would not have made this option feasible. To overcome the challenge of language barriers, potential master trainers were identified during the courses. These master trainers were invited to Singapore in 2019 to gain exposure to systems of hospital disaster preparedness, paramedical services, on-site disaster medical management and disaster exercises [27]. These master trainers, together with MOHP, Nepal, are planning to translate the course materials and have the course conducted in Nepalese in the future.

An on-site disaster exercise was also conducted in 2019 as part of a larger disaster drill to address one of the perceived shortcomings brought up during the two courses.

There is, currently, no compulsory national or regional disaster management course for relevant healthcare professionals in Nepal. The HOPE course initiated by the United States Agency for International Development and Office of United States Foreign Disaster Assistance in 1998 is nested under the Institute of Medicine, Tribhuvan University, for medical professionals who are self-motivated and interested [21]. This course has little emphasis on on-site disaster medical management. The pre- and post-test questions which demonstrated the highest degree of improvement were related to disaster on-site organization, coordination of medical services and the first-aid post, as well as triage. This suggests that the participants lacked knowledge but had benefitted in these aspects after the programme. The results also illustrate the need for the country to train larger segments of the healthcare sector in on-site disaster medical organization and management.

The participants fared worse on the question regarding disaster response agencies and their roles at the disaster site, especially during the second course. This could have been due to the condensation of the three-day course into two days with hurried or inadequate descriptions on roles of disaster agencies to achieve common understanding. Another possible explanation could be the underdevelopment of pre-hospital paramedical services in the country and the very small number of pre-hospital care participants in the courses [28]. These are likely to have resulted in healthcare workers being generally unaware of the major roles for ambulance services in disaster management, the current relative lack of inter-agency coordination and little involvement of the civilian healthcare sector in the initial phases of major disaster management [29]. A survey in 2006 showed that only 9.9% of patients arrived at a major hub hospital’s Emergency Department in Kathmandu via ambulances, as the public perceived that available ambulances were not equipped and did not have trained staff [30]. Additionally, the only well-organized ambulance service providing pre-hospital medical care in Nepal was a private agency called the Nepal Ambulance Service (NAS) which only began operations in 2011 and was, in 2017, facilitating only nine ambulance responses a day in the Kathmandu area, which had a population of 975,453 [31,32]. The national disaster medical response agency (HEOC) was also in its infancy, having only been established in 2012 [33]. When the 2015 earthquake struck, lack of leadership and coordination amongst medical staff were still key issues [3]. The WHO established a field office to assist in coordination of national and international humanitarian aid [34]. The underdevelopment of pre-hospital paramedical services and unfamiliarity with HEOC functions may have led to misunderstandings of terms such as “Site Incident Commander” by course participants. The tripartite agreement members have looked into the above issues and subsequently conducted a two-day HEOC networking training course in 2019, a half-day real-time exercise and advisory meetings with MOHP and NAS. The networking course explained the various structures in a disaster management environment at multiple levels of the organization and clarified terminologies such as incident commander, hospital commander, on-site medical commander, and national and provincial HEOC commander.

The general improvement in post-test results is not unexpected. Previous pre- and post-course tests in disaster management conducted in other environments have demonstrated improvement in knowledge and skills regardless of learning methods used [35,36,37,38].

### Limitations

Firstly, we did not ascertain who amongst the participants had undergone such previous training. Knowledge of previous training may have aided in their pre-test and post-test performance. This should be rectified during future courses.

Secondly, the evaluation method of a pre-test and post-test has its inherent disadvantages. Participants are expected to know more immediately after a workshop owing to learning momentum and recall. This potentially limits the usefulness of this education assessment tool.

However, our results showed that post-test results are not always better than pre-test results and that the pre-test and immediate post-test evaluation method may still have a role to play in assessing effectiveness of an education programme [39,40]. The post-test discussion also provided the participants instant feedback about their level of understanding of the particular topics. Employing a pre- and post-test methodology also does not assess long-term knowledge acquisition and retention, though this can be enhanced by repeating the same test after a period of time [41]. This may be an area of future study.

Thirdly, some participants took either the pre-test or post-test questionnaire but not both, owing to work commitments precluding their presence at the time of the administered tests. To account for these missing data, the mean imputation method was used. Whilst the drawback of the mean imputation method includes an underestimate of standard error, this would be less likely in this case owing to the randomness of the absence of the participants, and hence missing data points, across a range of healthcare occupations.

Fourthly, for course feedback, a three-point Likert scale was employed. A three-point Likert scale may have been too coarse, and this may have resulted in a potential loss of discriminative ability [42]. However, in our analysis, the use of the three-point Likert scale was still able to highlight notable differences, for example, that the condensed 2-day course’s delivery of content was clearly less well received than the 3-day course. Furthermore, a qualitative aspect was included, which internally validated the feedback scores.

Finally, we found that at times, tests with single questions may have not been enough to gauge knowledge on a particular area of the course conducted. However, when there is a consistent improvement in knowledge between the pre-course and post-course periods in multiple areas, it may be reasonable to assume that their knowledge in the course topics generally improved.

## 5. Conclusions

This study has shown that a three-day jointly prepared BOS-DMS course has the potential to improve knowledge and initial disaster management skills of the Nepalese medical community through interprofessional education.

## Figures and Tables

**Table 1 healthcare-12-01308-t001:** Contents from the course manual of Basic On-Site Disaster Medical Support.

Table of Contents
Chapter 1—Disaster Definitions Chapter 2—The Disaster Management Cycle Chapter 3—Types and Effects of Disasters in Nepal Chapter 4—The Need for Community-Based Emergency Response Chapter 5—Planning for Disasters Chapter 6—Disaster Emergency Services and their Roles Chapter 7—Organization of the Disaster Site Chapter 8—Medical Support for Disasters Chapter 9—Organization of Medical Services at a Disaster Site Chapter 10—Organization and Components of a First-Aid Post Chapter 11—Activation of Medical Services during a Disaster Chapter 12—Triage in Disasters Chapter 13—Communications in Disasters Chapter 14—Psychosocial Aspects of Disaster Chapter 15—Forensic Support in Disasters Chapter 16—Disaster-Site Medical Logistics Chapter 17—Hospital-Level Medical Support in Disasters Chapter 18—Coordination of Disaster Medical Services in a Community Chapter 19—Disaster-Site Medical Support in Special Disaster Situations: Floods, Earthquakes, Landslides, Fires

**Table 2 healthcare-12-01308-t002:** Professions of participants.

	Number of Participants (*n* = 135)
Nurses	54 (40.0%)
Physicians	48 (35.5%)
Allied Health Workers	12 (8.9%)
Administrators	16 (11.8%)
Ambulance crew	5 (3.7%)

**Table 4 healthcare-12-01308-t004:** Results of course feedback (figures in percentages).

	1st Course (2–4 May 2018)			2nd Course (26–27 November 2018)			Overall			
	Agree	Neutral	Disagree	Agree	Neutral	Disagree	Agree	Neutral	Disagree	*p*-Values
The programme has achieved its stated objectives.	94.6	5.4	0	90.9	9.1	0	92.4	7.6	0.0	0.698
Course materials/handouts were well designed/organized.	91.9	5.4	2.7	78.2	18.2	3.6	83.7	13.0	3.3	0.031
The contents were well covered.	94.6	5.4	0	74.5	23.6	1.8	82.6	16.3	1.1	0.013
The programme contained useful and practical exercises.	91.9	5.4	2.7	90.9	7.3	0	91.3	6.5	1.1	1.000
The duration of the course was appropriate.	67.6	29.7	2.7	47.3	23.6	18.2	55.4	26.1	12.0	0.086
Good knowledge of subject matter before the workshop.	27.0	43.2	29.7	41.8	43.6	14.5	35.9	43.5	20.7	0.118
Good knowledge of subject matter after the workshop.	91.9	5.4	2.7	89.1	10.9	0	90.2	8.7	1.1	0.736
Ability to apply knowledge and skills in course of work.	94.6	2.7	2.7	92.7	5.5	1.8	93.5	4.3	2.2	1.000
The trainer was well prepared for the course.	94.6	5.4	0	89.1	10.9	0	91.3	8.7	0.0	0.468
The trainer has adequate knowledge of the subject.	97.3	0	2.7	92.7	5.5	1.8	94.6	3.3	2.2	0.645
Trainer clear/concise in communicating ideas/concept.	86.5	13.5	0	65.5	25.5	9.1	73.9	20.7	5.4	0.030
Trainer encouraging and interacts well with participants	86.5	13.5	0	80.0	14.5	5.5	82.6	14.1	3.3	0.577
Trainer uses good examples/case studies/role play	91.9	5.4	2.7	96.4	3.6	0	94.6	4.3	1.1	0.388
The training environment was conducive for learning.	70.3	29.7	0	58.2	38.2	3.6	63.0	34.8	2.2	0.276
The use of visual aids is appropriate.	89.2	10.8	0	76.4	21.8	1.8	81.5	17.4	1.1	0.110
The class size was right for maximum participation.	64.9	27.0	8.1	81.8	16.4	1.8	75.0	20.7	4.3	0.052
Good quality of refreshments and food served.	62.2	35.1	2.7	69.1	29.1	1.8	66.3	31.5	2.2	0.509

## Data Availability

There are no additional data supporting this research, other than those already been presented and included in this manuscript.

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
