# Peer review of "Evaluation of an On-Site Disaster Medical Management Course in Nepal"

_healthcare, 2024, doi:10.3390/healthcare12131308_

Round 1

Reviewer 1 Report (Previous Reviewer 2)

Comments and Suggestions for Authors

I thank the authors for reviewing the manuscript and following the recommendations made by the reviewer. I think the paper has improved substantially.
I think two small aspects should be improved in the results section. The p-value that you must write is the one found (e.g. p=0.032) and not p<0.05. All p<0.05 should be removed and the exact value found entered. A second aspect is that the authors must decide the number of decimal places they want to put in the tables (p-value) and standardize it. There cannot be tables with two decimal places and others with four decimal places.

Author Response

  • Comment 1: I thank the authors for reviewing the manuscript and following the recommendations made by the reviewer. I think the paper has improved substantially.
  • Response 1: We thank the reviewer for his critique and recommendations.
  • Comment 2: I think two small aspects should be improved in the results section. The p-value that you must write is the one found (e.g. p=0.032) and not p<0.05. All p<0.05 should be removed and the exact value found entered. A second aspect is that the authors must decide the number of decimal places they want to put in the tables (p-value) and standardize it. There cannot be tables with two decimal places and others with four decimal places.
  • Response 2: We thank you for your comment. We have edited the p-values in the tables and in the main text. We have standardized them to 3 decimal places. There are values which are very small, for example: 0.00000123. For such values, we have standardized them to p<0.001.
  • Comment 3: 

Reviewer 2 Report (Previous Reviewer 3)

Comments and Suggestions for Authors

The topic is helpful and valuable in preparing disaster-related services through interprofessional education. A few suggestions below for further consideration.

·  May consider adding the sampling size calculation for the study, if available

·  The single sentence in 2.2 Course Participants (Lines 167-168) may be combined with the same session's second or third paragraph.

·    In the Discussion session, it seems the single sentence in Lines 327-328 is an incomplete sentence/paragraph. Please consider adding elaboration.

·     In the Discussion session, the last sentence in Lines 407-409 may be considered to have more elaboration.

·      In the Limitation session, the first sentence in Lines 413-414, may have more elaboration or suggestions for future courses.

·     In the Limitation session, Lines 419-421 sentence seems related to line 372.

·     A minor typo in Line 443, “three-point” instead of “three-pint”. 

Thank you.

Author Response

  • Comment 1: The topic is helpful and valuable in preparing disaster-related services through interprofessional education. A few suggestions below for further consideration. May consider adding the sampling size calculation for the study, if available.
  • Response 1: 
    • Thank you for your invaluable feedback and comments.
    • A sampling size calculation was not conducted, as the pre- and post-tests were given to the entire sample population.
  • Comment 2: The single sentence in 2.2 Course Participants (Lines 167-168) may be combined with the same session's second or third paragraph.
  • Response 2:
    • Lines 167-168: “The first course was conducted for three days from 2 to 4 May 2018 and the second for two days from 26 to 27 Nov 2018.”
    • We have separated the single sentence in 2.2 Course Participants to two parts. The first part was combined with the same session’s second paragraph and the second part was combined with the same session’s third paragraph. We hope that this will help with ease of readability.
    • The final edit will read like this: “The first course was conducted for three days from 2 to 4 May 2018. The course participants […]. The second course was condensed into a two-day program, and ran from 26 to 27 November 2018, at the request of local health authorities owing to time constraints of the participants.  The content […]”
  • Comment 3: In the Discussion session, it seems the single sentence in Lines 327-328 is an incomplete sentence/paragraph. Please consider adding elaboration.
  • Response 3: 
    • Lines 327-328: “The decrease in scores in the question on the role of disaster agencies was mainly observed in the results of the second course”
    • The single sentence in the manuscript was missing a period (full stop). We apologize and have added it in to form a complete sentence.
  • Comment 4: In the Discussion session, the last sentence in Lines 407-409 may be considered to have more elaboration.
  • Response 4: 
    • Lines 407-409: “The tripartite agreement has looked into the above issues and subsequently conducted a two-day HEOC networking training course in 2019, a half-day real-time exercise and advisory meetings with MOHP and NAS.”
    •  We have added the following sentence to elaborate what was taught during this course: “The networking course explained the various structures in a disaster management environment at multiple levels of the organization and clarified terminologies such as incident commander, hospital commander, on-site medical commander, national and provincial HEOC commander.”
  • Comment 5: In the Limitation session, the first sentence in Lines 413-414, may have more elaboration or suggestions for future courses.
  • Response 5:  We thank Reviewer 2 for this feedback. We have added more elaboration and a suggestion for future courses as follows: “Knowledge of previous training may have thrown light on their pre-test and post-test performance. This should be rectified during future courses”.
  • Comment 6: In the Limitation session, Lines 419-421 sentence seems related to line 372.
  • Response 6: We request Reviewer 2 to clarify which lines are being referred to. The lines that Reviewer 2 refers to cover different topics. Is there a typo error in Reviewer 2’s comment? Please advise.
  • Comment 7: A minor typo in Line 443, “three-point” instead of “three-pint”.
  • Response 7: We thank Reviewer 2 for pointing this out. We have corrected the typo error.

Reviewer 3 Report (New Reviewer)

Comments and Suggestions for Authors

Dear authors,

please find comments in the attached PDF.

Comments on the Quality of English Language

Author Response

  • Comment 1: Line 39: Please check, probably a typo
  • Response 1: We thank the reviewer for pointing this out. It was a typographical error. We have edited the sentence to “The last major earthquake was in April 2015 with a magnitude of 7.6 recorded by Nepal’s National Seismological Centre was followed by more than 300 aftershocks of magnitude greater than 4.0.”

  • Comment 2: Line 148: The authors mean UNDRR? Please correct into “Reduction”
  • Response 2: Thank you for pointing this out. We have edited the phrase to “United Nations Disaster Risk Reduction (UNDRR).”

  • Comment 3: Line 173: “and”
  • Response 3: It was a typographical error. We have edited “ad” to “and”

  • Comment 4: Line 174: “Doctors selected” – change into Selected Doctors
  • Response 4: We have edited to “Selected doctors were […]”

  • Comment 5: Line 176: “Nurses and administrators selected” – As above
  • Response 5: We have edited to “Selected nurses and administrators were […]”

  • Comment 6: Line 220: Please clarify this sentence. Maybe the authors mean “Fisher’s exact test was used to compare…where relevant…; a p- value of…was deemed significant.”
  • Response 6: Thank you for your feedback. We have edited the sentence to “For the feedback portion, descriptive analysis was carried out and presented as a percentage of participants who agreed, were neutral or disagreed with each statement. Fisher’s exact test was used to compare between the three categorical variables. A p value < 0.050 would suggest a statistically significant finding for that statement.”

  • Comment 7: Line 234: Table 2 - Regarding institutions: such data are not commented in the results. Therefore, the authors should either comment them in results + discussion or remove from table 2 to supplementary materials.
  • Response 7:
    • Thank you for these comments. We have removed the second half of the table showing organization affiliations to create Supplementary Table 2, as these were not mentioned in the results or discussion.
    • We have retained the first half of the table showing breakdown by professions for the course participants, as we have commented on scores stratified by doctors and nurses from line 259 to 261: “For the doctors attending the courses the mean test scores after imputation increased from 5.42 ± 1.77 to 7.63 ± 2.04 (p < 0.001). For the nurses, the scores increased from 4.11 ± 1.22 to 6.30 ± 2.27 (p < 0.001).”

  • Comment 8: Line 311: Discussion – Put a comment at the beginning: e.g., is this the first study in the field of Nepal? Just to clarify the value of the paper to the reader.
  • Response 8: We thank the Reviewer for the suggestion. We have added in the following sentence at the start of Discussion. “This was the first before-and-after study comparing knowledge scores in disaster medical training in Nepal. In 2017, a systematic […]”

  • Comment 9: Line 377: Demonstrates – moderate to suggests
  • Response 9: Thank you for the suggestion. We have changed “demonstrates” to “suggests”.

  • Comment 10: Line 386: Pre-hospital – maybe the authors could mention that only a few participants from EMS participated in this course, which is a limitation.
  • Response 10: We thank the Reviewer for this suggestion. We have edited the sentence to include this observation. The sentence now reads, “Another possible explanation could be the underdevelopment of pre-hospital paramedical services in the country and the very small number of prehospital care participants in the courses.”

Reviewer 4 Report (New Reviewer)

Comments and Suggestions for Authors

Overview: The authors developed and presented a Disaster Management Course for health care providers in Nepal. They evaluated participant gain in knowledge from two separate courses (pre- versus post-test scores) and provided suggestions to others administering such courses. The document I received for review included redlined changes presumably made to a prior submission.

Abstract: The sentence on line 28, “Unforeseen scheduling conflicts also affected the effectiveness of course delivery” could be deleted as it is duplicative of the information in the concluding sentence recommendations.

Introduction: The authors summarize the need for a disaster management course tailored for the needs of Nepal. It would be helpful to indicate whether the focus would be on mass casualty care related to an earthquake or other destructive event (e.g., bomb) in which health care facilities may be compromised versus other issues such as power or water disruption, a mass chemical exposure, or overwhelming infectious disease exposure. Although an all-cause disaster management approach could be used for such training, focusing on the management of mass casualty care when health care facilities may be compromised would be most beneficial.

Materials & Methods: Again, the focus of the disaster management course should be made more explicit. Specifically, was the focus on "all cause disasters" or on mass casualties management with disruptions in medical care facility operations? If addressing disruptions in medical care facilities operations was incorporated in the training, how were hospital administrators engaged in command center discussions/operations with clinicians?

Results: Matched pre- versus post-test scores are reported for both courses in those learners without a missing exam. Similar mean score changes were noted after imputation of missing test results and inclusion of other learners. Results were similar for nurses as for physicians and for the first (3 day) versus second (2 day) course. As the participants spoke of the application of knowledge during the workshop, additional information should be provided regarding the types of scenarios used to provide practical experience in trauma triage, setting hospital operations/resource priorities, addressing any hospital facilities/operations impairments, etc.

Discussion: The sharing of logistical challenges in course delivery is a strength of the paper.

Limitations: Assessments were done at the beginning (pre-, i.e., before formal training) and end (post-, i.e., immediately after formal training). A delayed assessment to determine knowledge retention and/or the ability to build upon knowledge from the course would have enhanced the assessment of the course. The information in this section is helpful, but probably should be more concisely stated.

Addendum: Will greater online details of the course be made available to interested readers via a web-link?

Comments on the Quality of English Language

Minor syntax changes by editors would help the readability. No specific additional recommendations. 

Author Response

  • Comment 1: Abstract: The sentence on line 28, “Unforeseen scheduling conflicts also affected the effectiveness of course delivery” could be deleted as it is duplicative of the information in the concluding sentence recommendations.
  • Response 1: We thank Reviewer 4 for the suggestion. We have deleted that sentence.

  • Comment 2: Introduction: The authors summarize the need for a disaster management course tailored for the needs of Nepal. It would be helpful to indicate whether the focus would be on mass casualty care related to an earthquake or other destructive event (e.g., bomb) in which health care facilities may be compromised versus other issues such as power or water disruption, a mass chemical exposure, or overwhelming infectious disease exposure. Although an all-cause disaster management approach could be used for such training, focusing on the management of mass casualty care when health care facilities may be compromised would be most beneficial.
  • Response 2: We thank Reviewer 4 for this suggestion. We have added the words “which adopted an all-cause disaster management approach” to the sentence from lines 96-100 which will now read as follows: “The objective of this course, which adopted an all-cause disaster management approach, was to enable local healthcare institutions and healthcare workers in the vicinity of a disaster to mobilize their resources to provide coordinated medical assistance at the local disaster site as soon as possible after the onset of the incident as is recently being arranged in a number of other countries”. In the course, after teaching the all-cause disaster management approach, we presented and discussed scenarios and on-site management in the event of floods, fires, earthquakes and landslides and practical scenarios such as aeroplane crashes and chemical incidents. For all these we used the same approach and discussed the special variations that the participants would need to adopt for the different scenarios. We also taught and discussed organisation of hospitals in different situations, including when hospital structures and services were disrupted and when dealing with chemical disasters. However, some of these details are not described in the main manuscript and would be seen in a table on the detailed training programme which had earlier been placed as a supplementary table, instead of in the main manuscript, at the request of other Reviewers.

  • Comment 3: Materials & Methods: Again, the focus of the disaster management course should be made more explicit. Specifically, was the focus on "all cause disasters" or on mass casualties management with disruptions in medical care facility operations? If addressing disruptions in medical care facilities operations was incorporated in the training, how were hospital administrators engaged in command center discussions/operations with clinicians?
  • Response 3: We thank Reviewer 4 for this query. This course for which this manuscript was written focused on On-site Disaster Management. Though some portion of this course addressed hospital organisation and hospital operations, the bulk of the course covered medical support being provided at disaster sites. Separate courses on Hospital management in Disasters were also organised and conducted by us in Nepal that addressed in greater detail disruptions to medical facility operations and how hospital administrators were to engage in these. However, the focus of this manuscript is on the courses we conducted on disaster-site medical support, not on those that covered hospital management or coordination of medical facilities in a community. We request the Reviewer’s understanding on this.
  • Comment 4: Results: Matched pre- versus post-test scores are reported for both courses in those learners without a missing exam. Similar mean score changes were noted after imputation of missing test results and inclusion of other learners. Results were similar for nurses as for physicians and for the first (3 day) versus second (2 day) course. As the participants spoke of the application of knowledge during the workshop, additional information should be provided regarding the types of scenarios used to provide practical experience in trauma triage, setting hospital operations/resource priorities, addressing any hospital facilities/operations impairments, etc.
  • Response 4: We thank Reviewer 4 for these comments. We have, therefore, amended paragraph 2.1 of the Results section of the manuscript to incorporate the following:

“Each workgroup consisted of a mix of eight to ten participants, with a SingHealth emergency physician as facilitator to guide discussion, practice and answer queries. Every workgroup was expected to be able to address and solve near-realistic scenarios.

The practical scenarios used to provide practical experience were:

  • Multiple trauma triage scenarios that may be expected in mass casualty situations
  • Medical support planning for on-site medical support in an aircrash situation
  • Practising activation and dispatch of medical teams to a disaster site
  • Practice setting up a first-aid post and basic simulation using casualty cards
  • Practical training on disaster-site communications at a first-aid post
  • Practising command and control of a first-aid post at a disaster site
  • Practical training on scenarios that may be encountered at the Red, Yellow and Green areas and at the Ambulance point of a first-aid post
  • Practical training on medical support at disaster-sites during floods, landslides and fires
  • Practical training on decontamination techniques during chemical disasters
  • Small group discussions on how to conduct disaster site exercises.

These scenarios were relevant to the local setting and addressed lessons covered in the lectures. Presentations by each workgroup to the whole class provided opportunities for larger group discussions and sharing of personal experiences.

  • Comment 5: Discussion: The sharing of logistical challenges in course delivery is a strength of the paper.
  • Response 5: Thank you

  • Comment 6: Limitations: Assessments were done at the beginning (pre-, i.e., before formal training) and end (post-, i.e., immediately after formal training). A delayed assessment to determine knowledge retention and/or the ability to build upon knowledge from the course would have enhanced the assessment of the course. The information in this section is helpful, but probably should be more concisely stated.
  • Response 6: We thank Reviewer 4 for this comment. We have revised this portion of the Limitations section and shortened it significantly.

  • Comment 7: Addendum: Will greater online details of the course be made available to interested readers via a web-link?
  • Response 7: Yes, in the Supplementary tables and the access to the database that will be made available via a web-accessed link.

This manuscript is a resubmission of an earlier submission. The following is a list of the peer review reports and author responses from that submission.

Round 1

Reviewer 1 Report

Comments and Suggestions for Authors

Excellent paper and more importantly excellent work - we need more of this! A few questions

Where the disaster definitions, terminology and abbreviations based on the UNDRR definitions https://www.undrr.org/drr-glossary/terminologyor were they from another recognised body? Please state.

Which triage system was used?

At some stage you might like to connect with National Disaster Life Support Foundation, Inc. https://www.ndlsf.org/adls. You have done wonderful work and we really need to have an international focus when it comes to disaster management

Reviewer 2 Report

Comments and Suggestions for Authors

I thank the authors for the opportunity to read this interesting manuscript.

First of all, I think the title may be misleading. It is not about the evaluation of a program but rather a course and this should be reflected in the title.

The introduction should be improved. Similar courses and programs carried out in other environments are not described, nor is their effectiveness, nor is bibliography provided on the subject.

The entire introductory part that refers to how the course being evaluated has been designed should be positioned in materials and methods. Table 2 does not provide relevant data to the paper and should be positioned, in any case, as supplementary material.

Authors should provide the exact question that was asked (in a table) and not just the "topic".

The methodology is unclear. The authors should describe how the selection of students was carried out, and in an initial table describe age, sex, previous training, etc. Given the diversity of students, do the authors present the same objectives for doctors and nurses as for ambulance crews? Information on the results should be given according to the profession of the participants.

As the authors indicate, not all students answered the tests and perhaps one student who did answer did not answer all the questions. You should put the "n" in each question.

The authors do not indicate in the methodology what significance they propose for their study (p<0.05?)

Table 4 is confusing, the "n" of each question is not provided. Given that it is before-after and only 87 participants answered the posttest, it is understood that the table refers to 87 participants, is that correct? The standard deviations should be provided in the table. Does the "p" value of the table refer to the first, second or third columns?

The authors indicate (line 144) that 113 participants increased their score from 4.24 to 6.55. How is this possible if only 87 participants participated in the subsequent test?

It strikes me that the authors assume that by answering well a single question about a given area (and that had already been known by the participants in the pretest) they consider that the participants have significantly increased their knowledge in that area.

It seems to me that the discussion does not address in depth the comparison with similar experiences carried out in other environments.

The conclusion should be derived solely from the results, however the authors present many personal reflections, which, although they may be true, should not be in the conclusions, but in the discussion.

Reviewer 3 Report

Comments and Suggestions for Authors

Thanks for the invitation to review this manuscript. The program's topic is meaningful and helpful in preparing the healthcare sector's disaster-related services through interprofessional education. The following are some comments and suggestions for consideration.

· It would be better to consistently mention the same program's name throughout the abstract and the article sessions.

·   The introduction mainly described Nepal's earthquakes in April 2015 and the local need to develop a disaster management program in Nepal. More comprehensive and updated literature or evidence is suggested to support the research study's needs and significance. The tables presented in the introduction could be simplified and concise.  

· Careful consideration and appropriate research design and methodology are recommended when conducting a research study.

· The study design, sampling size, and sampling method could be clearly mentioned. Hospital senior management and local community authorities recommended or nominated the participants. Are there any inclusion or exclusion criteria for the subjects recruited, e.g., able to read and understand English?

·  The introduction session described the program's development and contents. However, the study was not reported in a timely manner as the two courses were conducted in May 2018 and November 2018, more than five and a half years ago.

·   Though explanations of discrepancies in the duration of the two courses were given, the two courses are best to be conducted at a standard pace (3 days and 2 days).

·   The reliability of the survey for data collection was not mentioned. Also, how to collect and analyze and what kinds of qualitative feedback were collected from the participants.

·    The survey questions can be clearly presented in Table 4 for the readers' information.

Thank you.

Reviewer 4 Report

Comments and Suggestions for Authors

Please confirm the "instructions for authors". Your paper has not been properly formatted and we recommend that you re-format it and re-submit it.